# Assessment of the Tissue Resident Memory Cells in Lesional Skin of Patients with Psoriasis and in Healthy Skin of Healthy Volunteers

**DOI:** 10.3390/ijerph182111251

**Published:** 2021-10-26

**Authors:** Marta Kasprowicz-Furmańczyk, Joanna Czerwińska, Waldemar Placek, Agnieszka Owczarczyk-Saczonek

**Affiliations:** Department of Dermatology, Sexually Transmitted Diseases and Clinical Immunology, University of Warmia and Mazury in Olsztyn, Oczapowskiego 2, 10-719 Olsztyn, Poland; joannaj061@gmail.com (J.C.); w.placek@wp.pl (W.P.); aganek@wp.pl (A.O.-S.)

**Keywords:** psoriasis, tissue resident memory cells, TRM, immune memory

## Abstract

Background: In the course of plaque psoriasis, tissue resident memory cells (TRM) are responsible for the phenomenon of “immune memory” of lesions, i.e., the appearance of recurrences of lesions in the same location, as well as Koebner phenomenon. We present results determining the location and amount of TRM in psoriatic lesions in patients suffering from plaque psoriasis, as well as an analysis of the relationship between TRM markers expression and the duration and severity of the disease. Methods: TRM markers (CD4, CD8, CD103, CD69, CD49, CXCR6) and tissue expression of cytokines (IL-17, IL-22) in the lesional psoriatic skin of 32 patients compared with 10 healthy skin samples were evaluated by immunohistochemistry. Results: The presence of TRM markers in both the epidermis and skin with psoriatic eruptions was demonstrated in much higher amounts compared with the skin of healthy volunteers. A significant positive relationship was demonstrated between the expression of TRM markers in patients with plaque psoriasis and the duration of skin lesions. There was no relationship between the amount of TRM and the severity of plaque psoriasis. Conclusions: A thorough understanding of the mechanisms responsible for the development and relapse of plaque psoriasis may contribute to the implementation of more effective therapies.

## 1. Introduction

In healthy skin, the number of T cells occupying the skin is almost twice as high as in peripheral blood, and most of them are effector memory cells (TEM), formed after antigen exposure from naïve T cells [1]. Effector memory cells have an immune defense function. Most of these cells live briefly and die after an immune response, but some morph into memory T cells. Central memory T cells (TCM) move through lymphoid tissues while TEM circulate in peripheral tissues [2,3]. TRM are subset of memory T cells that persist in non-lymphatic peripheral tissues for a long time without recirculation in the blood, thus providing the first line of adaptive cellular defense [2].

The role of TRM in the body’s immune defense is not yet fully understood, but it is likely that these cells can activate both the innate and adaptive immune systems [4]. In addition to protective functions, more and more studies indicate that inappropriate TRM activation may be involved in pathological conditions such as vitiligo, alopecia areata, psoriasis, skin T-cell lymphoma, or melanoma [1,2,5].

Psoriasis is currently regarded as a Th1/Th17/Th22-mediated systemic inflammatory disease characterized by several comorbidities encompassing respiratory [6,7], cardiologic [8,9], and gastrointestinal systems [10].

In the course of plaque psoriasis, TRM are responsible for the phenomenon of “immune memory” of lesions, i.e., the appearance of recurrences of lesions in the same location [11]. Moreover, after the psoriatic plaques have disappeared, signs of inflammation in the form of TRM cells can still be found in healthy skin. They appear to be able to initiate an inflammatory cascade, causing the recurrence of psoriatic plaques [4]. There are two main types of TRM: CD8 +, which is abundant in the psoriatic epidermis, and TRM CD4 +, which are localized near the vessels in the dermis and have a high proliferative potential [12]. CD8 + epidermal TRM mainly express the antigens CD103, CD69, and CD49a. Other molecules that distinguish TRM cells from other circulating memory T cell types have also been described, including CXCR3, CXCR6, and CD101 [1]. The particular pathogenicity of TRM CD8 + in psoriasis is evidenced by the fact that they express the receptor for IL-23 and can produce pro-inflammatory IL-17 and IL-22 in the skin, even many months after the lesion has resolved [4,13].

An interesting issue in the context of TRM is also the Koebner phenomenon. It turns out that physical stimuli can lead to the accumulation of memory T cells and even their reactivation [12]. The mechanism of this phenomenon has not been fully elucidated; however, numerous studies have shown that intact skin in inflammatory dermatosis differs markedly from normal skin in terms of increased expression of genes related to the immune system and certain T cell-related cytokines and adhesion molecules [12,14,15]. The relationship between the accumulation of TRM cells in nonlesional skin after the initiating stimuli and the Koebner effect is highly probable.

In this article, we present results describing the location and amount of TRM in psoriatic lesions in patients suffering from plaque psoriasis, as well as an analysis of the relationship between TRM markers expression and the duration and severity of the disease.

## 2. Methods

### 2.1. Study Group

The study group included 32 patients (26 men and 6 women) with plaque psoriasis, without psoriatic arthritis, who were untreated for at least 4 weeks and who were treated at the outpatient clinic at the Department of Dermatology, Sexually Transmitted Diseases, and Clinical Immunology in Olsztyn. The patients followed a normal diet (omnivores), and they were not addicted to alcohol. Patients with chronic and acute inflammatory diseases and dermatoses other than psoriasis, neoplastic diseases, previous cardiovascular complications, heart, kidney and liver failure were excluded. The average duration of the disease in women was 15 years, in men 14.4 years. In the study group, we also assessed the severity of the disease using the following scales: PASI, BSA, DLQI. The control group consisted of healthy volunteers (*n* = 10) with no personal or family history of psoriasis and with no concomitant autoimmune and inflammatory diseases. Skin samples from psoriasis patients were collected from psoriatic lesions that recurred at the same sites after the end of previous treatment. Healthy volunteer skin samples were obtained from surgical wastes obtained after removal of pigmentary lesions located on the trunk or limbs.

### 2.2. Clinical Samples

Assessment of TRM in biopsy specimens was performed by immunofluorescence method. We obtained one 4 mm punch biopsy per patient from the center of the psoriatic plaque and one from healthy volunteers (healthy skin), using local anesthesia (1% lignocaine). Tissue samples (lesional skin from psoriatic patients and non-lesional skin from healthy individuals) were cut into 5 μm thick sections in the CM3050 cryostat system (Leica, Buffalo Grove, IL, USA) and mounted onto glass slides coated with poly-L-lysine (Menzel-Glaser, Braunschweig, Germany). Frozen sections of the examined tissues were thawed to room temperature and fixed in acetone. After rinsing in 0.01 M PBS, they were incubated with 2.5% normal horse serum for 30 min at room temperature (Vector Laboratories, Burlingame, CA, USA) to decrease nonspecific binding. Then, they were incubated at 4 °C overnight with mouse anti-CD8 and CD4 or rabbit anti-CD103, CD69, CD49, CXCR6, IL-17, and Il22 polyclonal antibody (1:50; Merck Millipore, Billerica, Massachusetts). On the following day, the sections were washed 3 times in PBS and incubated for 30 min with secondary horse anti-mouse/rabbit antibodies (commercially diluted; ImmPRESS Universal reagent Anti-Mouse/Rabbit Ig; Vector Laboratories, Burlingame, CA, USA). In negative controls, 0.01 M PBS was applied instead of primary and/or secondary antibodies (three different controls). To present histology of the tissue section, hematoxylin was used (Figure 1). The immunohistochemical specimens were viewed under a fluorescent microscope (CH30/CH40; Olympus, Tokyo, Japan).

The results were processed statistically by non-parametric Mann–Whitney U test. The results are expressed as means ± SEM. Correlation between proteins was analyzed by Spearman’s test (*p* < 0.05). All calculations were performed using the Statistica program, release 13 (Statsoft, Inc., Tulsa, OK, USA). Differences were regarded as statistically significant at *p* < 0.05.

The study was approved by the Bioethical Committee of the Warmia and Mazury University in Olsztyn on 29.04.2020 (Resolution 24/2020). Informed consent was obtained from each patient enrolled in the study.

## 3. Results

### 3.1. Immunoreactive Area and Skin Localization of TRM Markers in Lesional Skin in Comparison with Healthy Control

The immunoreactive area (%) of CD8, CD4, CD103, CD69, CD49, CXCR6, IL-17, and IL22 was assessed in the affected skin (*n* = 32, dermis and epidermis) compared with the healthy control (*n* = 10, dermis and epidermis) (Table 1; Figure 2). The relationship between the expression of TRM markers and the duration and severity of the disease was also analyzed.

In the case of CD8 (Figure 3A) and CD69 (Figure 3E), a statistically significant difference in their amounts was demonstrated between the epidermis and the dermis, both in the control group and in patients with psoriasis (*p* < 0.05). In both cases, a much larger immunoreactive surface was located in the epidermis. Moreover, in psoriatic patients, an increase in the immunoreactive area of CD8 and CD69 in the epidermis was found, while it did not change significantly in the dermis.

There was a significant difference between the amount of CD4 + in the epidermis and the dermis in the control group (*p* < 0.05) and in people with psoriasis (*p* < 0.05)—the amount was higher in the dermis. Additionally, in the skin of psoriatic patients, an increase in the immunoreactive area of CD4 was found in both the epidermis and skin.

CD103 (Figure 3C) and CD49 (Figure 3D) also showed differences between the epidermis and dermis in the control group (*p* < 0.05) and in people with psoriasis (*p* < 0.05)—there was more in the epidermis. Additionally, in psoriasis patients, a significant increase in the immunoreactive area of CD103 was demonstrated in both the epidermis and the dermis.

In the case of CXCR6 in the control group, there was no difference between the epidermis and the dermis, but there was a clear increase in the area of CXCR6 in the group of patients in two layers of the skin, and in addition, the level was higher in the epidermis compared with the dermis.

In IL17 (Figure 3G) and IL22 (Figure 3H), the levels of the epidermis and dermis in both groups did not differ, but there was a marked increase in area in both the epidermis and dermis of the psoriasis group.

Correlation analysis in the control group showed only two relationships. A relationship was observed between changes in CD8 and CD103 expression (r = 0.62) and between CD8 and CD49 (r = 0.69).

There were many more of these correlations in the diseased tissue. Increases in the expression of most of the markers—CD8, CD4, CD103, CD49, CD69, CXCR6—were correlated with each other (the correlation coefficient “r” is presented in Table 2, *p* < 0.05). Interestingly, negative correlations were observed between CD4 and all other factors, which were related to the different nature of changes in these markers between the individual layers of the skin sections: epidermis and dermis (marked increase in CD4 in the skin). In contrast, IL17 was only correlated with IL22 (r = 0.45, *p* < 0.05).

### 3.2. Correlations between TRM Markers and the Duration of Psoriatic Lesions

In our study, we also analyzed the relationship between marker expression and disease duration, showing a clear relationship between these variables (Table 3). The longer the course of the disease, the higher the expression of TRM markers was observed (time vs. marker, *p* < 0.05).

### 3.3. Correlations between the Analyzed Variables: Time, Intensity, and Expression of TRM Markers

The evaluation of the PASI, BSA, and DLQI scales allowed for the analysis of the correlation between the obtained values, disease duration, and the expression of the analyzed markers. There was no correlation between the severity of lesions and the time of their occurrence (PASI, BSA, DLQI vs. time). The expression of the analyzed substances was not dependent on the intensity of skin lesions (marker vs. PASI, BSA, DLQI). Only a statistically significant relationship was observed between the scales themselves (the “r” values are presented in Table 4).

## 4. Discussion

CD8 + TRM T cells are plentifully present in the psoriatic epidermis, and their number correlates with the thickness of the epidermis [1,16], while CD4 + TRM preferentially inhabit the dermis [4]. The key skin TRM surface markers are CD69, integrin αE (CD103), integrin α1 (CD49a), and CXCR6 [17]. In our group of 32 psoriasis patients, there was statistically significant increase in the immunoreactive area of CD8 in the epidermis (Figure 3A), CD4 in the dermis and, to a lesser extent, in the epidermis (Figure 3B) compared to the control group. It seems that the accumulation of epidermal CD8 + cells induces keratinocyte hyperproliferation as well as papillomatosis (increase in CD8 + concurrent to the intensity of Ki67 staining in keratinocytes) [4,13]. In psoriatic patients, epidermal CD8 + TRM cells express the CLA antigen, CCR6, CD103, and IL-23R and produce IL-17A upon ex vivo stimulation, which proves their pathogenicity. The epidermis can also be occupied by CD4 + CD103 + TRM producing IL-22 upon stimulation [1,4]. Both IL-17 and IL-22 are known to play an important role in the pathogenesis of psoriasis. Our studies also confirm a significant increase in IL-17 (Figure 3G) and IL-22 (Figure 3H) levels in both the epidermis and dermis of psoriatic patients. The amounts of these interleukins turned out to be significantly higher than in the skin and epidermis of healthy patients (Table 1). High amounts of pro-inflammatory IL-17 and IL-22 persist for months after the lesion has cleared, and their amount is correlated with the duration of the disease (Table 3).

CD103 is a ligand for E-cadherin, an adhesion molecule expressed by epithelial cells in barrier tissues [18]. Its expression is most pronounced on epidermal CD4 + and CD8 + TRM cells because it allows TRM to bind to E-cadherin, widely expressed by epithelial cells [19]. This determines the adhesion and also promotes local retention of TRM. In addition to adhesion, CD103 + TRM are involved in cytotoxicity through exocytosis of cytolytic granules. Research shows that the lack of CD103 results in a reduction in but not a removal of the TRM population. The above data prove that the presence of TRM in tissues is not only determined by binding to epidermal cells [20]. In the control and study groups, we showed a significant difference between the amount of CD103 in the epidermis and dermis—a much larger immunoreactive area in the epidermis (Figure 2C). In one study assessing phenotypic features of TRM in psoriatic patients without lesions and healthy skin, an immunofluorescence study revealed that TRM CD103 + CD8 was dominant in the epidermis compared with the dermis of the healthy control group [21]. We obtained similar results among our psoriasis patients—CD103 and CD49 were found in significantly greater amounts, mainly in the epidermis, but also in the dermis when compared with healthy subjects (Table 1).

CD49a is the α1β1 integrin receptor α subunit, expressed only on the epidermal CD8 T cells and has been confirmed to be a marker of TRM in the epidermis. CD49a binds to type IV collagen, which is located in the basement membrane zone and also has significant effect on cell migration along the collagen. It is intended to confine CD49a-expressing TRM cells to the epidermis. It is worth noting that the expression of CD49a determines the cytokine production profile of TRM cells [20]. CD8 + CD49a + TRM cells located in the epidermis produce perforin and IFN-γ—crucial in fighting viral infections. In our control group, it turned out that a significantly greater amount of CD49 was found in the epidermis compared with the dermis 10.66 ± 3.23% vs. 1.55 ± 0.34% (Figure 2D). The marker CD49a was confirmed to identify two distinct populations of CD8 + CD103 + TRM cells. CD8 + CD49a cells in psoriasis mediate disease by producing interleukin-17 (IL-17) [22,23]. However, blockade of IL-17 may require long-term administration because it blocks a molecule made by TRM and does not target TRM itself. In a clinical study of psoriatic patients (*n* = 10) treated with secukinumab for 24 weeks, Fujiyama et al. showed that although there was a significant decrease in the number of CD8 + CD103 + cells in the affected skin, there was only a slight decrease in the number of CD8 + CD103 + CD49- cells, suggesting that TRM cells are still preserved [23,24].

Another key TRM marker is CD69-a glycoprotein, which is involved in distinguishing T lymphocytes in tissues from those in the circulation and is responsible for their colonization in tissues, inhibiting their recirculation [4]. CD69 contributes to the downregulation of the sphingosine 1 phosphate receptor (S1P1) by inhibiting its expression. It allows CD69 + TRM to be distinguished from TEM (effector memory), which are CD69-negative. S1P1-expressing T cells are directed from the tissue to the lymph nodes and subsequently to the blood, depending on S1P gradients, the expression levels of which vary with localization (low in tissues, average in lymph nodes, high in blood) [25]. TRM expressing CD69 do not express S1P1, and therefore, they do not migrate from peripheral tissues. Accordingly, CD69 blocks tissue exit mediated by the sphingosine-1-phosphate-1 receptor (S1PR1). However, the lack of CD69 only results in a reduction in, not a complete disappearance of, the TRM population, as is the case with CD103 [20]. In the case of the tested control group, we found a significant difference in the amount of CD69 between the epidermis and the dermis in the control group, in favor of the epidermis (13.89 ± 2.51% vs. 6.39 ± 0.69%) (Figure 3E). Moreover, in an immunohistochemical study of skin lesions in patients with plaque psoriasis, we found a significant increase in the immunoreactive area of CD69 in the epidermis (25.24 ± 1.42%), but these values did not change in the dermis (7.94 ± 1.13%).

CXCR6 is expressed on human skin TRM cells, and the chemokine CXCL16, the ligand CXCR6, is expressed on epidermal keratinocytes and can be released as a chemoattractant. CXCR6-deficient T cells have a low ability to form TRM cells in the skin [1]. In the control group, we found no differences between the immunoreactive area (%) in the epidermis and the dermis—they remained at a very low level in both locations (2.24 ± 0.41% and 2.05 ± 0.33%, respectively) (Figure 3F). However, we observed a marked increase in CXCR6 in the group of psoriatic patients in both layers—higher in the epidermis (12.23 ± 1.04% vs. 7.34 ± 0.78%). These studies clearly indicate that CXCL16–CXCR6 interactions mediate the colonization of T lymphocytes in human skin, and thus contribute to the pathogenesis of psoriasis.

Keratinocytes in lesion-free skin that have never experienced disease in psoriatic patients are also involved in the accumulation of TRM. Undamaged keratinocytes tend to upregulate CCL20 expression upon cytokine stimulation, leading to the migration of IL-17A-producing T cells expressing CCR6 [20]. These data suggest that a CD8 TRM population with an IL-17A-producing profile is constructed in the skin of psoriasis patients before disease onset in response to recruiting signals such as CCL20 and ICAM-1 and cytokines such as IL-23 and IL-15 and contributes to the appearance of psoriasis lesions in the future [20].

Complete TRM suppression appears to be required for complete disease remission. Unfortunately, TRM cells are long-lived and resistant to destructive factors and apoptosis, and their elimination seems impossible [26]. Interestingly, after the psoriatic lesions have disappeared, TRMs can still produce IL-17A, and proper therapy only inhibits their activity [4].

Our analysis of the relationship between the expression of TRM markers and the duration of the disease showed a clear relationship between these variables. Marker expression in the study group was significantly dependent on the time of skin lesions (time vs. marker, *p* < 0.05). Longevity of TRM and their unquestionable ability to accumulate in the skin of psoriatic patients may explain the rapid recurrence of psoriatic lesions at the same site after the causative agent has been triggered.

We found no relationship between the severity of lesions and the time of their occurrence (PASI, BSA, DLQI vs. time). It is interesting that the expression of the analyzed markers was also not dependent on the intensity of skin lesions (marker vs. PASI, BSA, DLQI).

So far, there have been few publications assessing the effect of topical or systemic treatment on the amount of TRM in the skin of patients suffering from plaque psoriasis [27]. Cheuk et al. investigated the effect of nb-UVB, iTNF-α, or IL-12/23 therapy on TRMs. They found the presence of epidermal TRM cells at the site of resolved psoriatic papules and that these cells are capable of producing cytokines that play a key role in the pathogenesis of psoriasis [28]. However, ultraviolet irradiation seems to diminish the number of IL-17A-producing T cells in skin [29], and this T cell fraction includes TRM. Khalil et al. showed that CD8 + cells expressing CD69 remain in healed skin even several months after successful treatment with methotrexate, and CD4 + cells producing IL-22 and CD8 + producing IL-17 also remain in the epidermis of healed lesions [2]. Mehta et al. using flow cytometry, examined the profile of T lymphocytes in the same patients’ lesions before and during treatment with guselkumab (IL-23 blocker) or secukinumab (IL-17A blocker). Inflammatory dendritic cells and CD4 + CD49a-CD103- T cells were reduced with both treatments. In contrast, guselkumab decreased memory T cells while maintaining regulatory T cells, contrary to secukinumab. Neither drug modified the frequency of IL-17A + IL 17F +/− CD4 + or CD8 + T cells [30]. In the case of topical treatment, vitamin D analogues and corticosteroids reduce the lesional IL-17A-producing TRM, possibly also reducing pathogenic TRM [31,32,33].

Taken together, these results indicate that although we achieve clinical resolution of psoriatic lesions, essential inflammation, as defined by the expression of key cytokines and chemokines, is not completely resolved in all psoriatic lesions and can be stimulated to initiate relapse. Undoubtedly, the issue of the influence of therapy on the amount of TRM requires further research.

## 5. Conclusions

We have demonstrated an increased number of TRM cells and their markers in the epidermis and dermis of psoriatic eruptions, compared with the skin of healthy volunteers. An interesting and important conclusion is that there is a clear positive relationship between the expression of TRM markers in patients with plaque psoriasis and the duration of skin lesions and the disease. This proves the unquestionable ability of TRM to accumulate in the skin of patients suffering from psoriasis. However, the amount of TRM is not affected by the intensity of skin lesions (assessed according to PASI, BSA scales).

Targeting TRM cells appears to be a new potential therapeutic strategy for reducing psoriasis flare-ups. However, due to the still low number of reports on the effect of local and general treatment on the amount of TRM in the epidermis and dermis of patients suffering from psoriasis, further research is required in relation to this issue. A thorough understanding of the mechanisms responsible for the development and relapse of plaque psoriasis may contribute to the implementation of more effective therapies.

## Figures and Tables

**Figure 1 ijerph-18-11251-f001:**
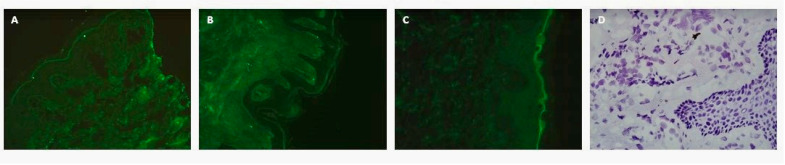
The representative photos of negative controls: without the primary (**A**), secondary (**B**), both types of antibodies (**C**), and staining with hematoxylin (**D**). Magnification: 500×.

**Figure 2 ijerph-18-11251-f002:**
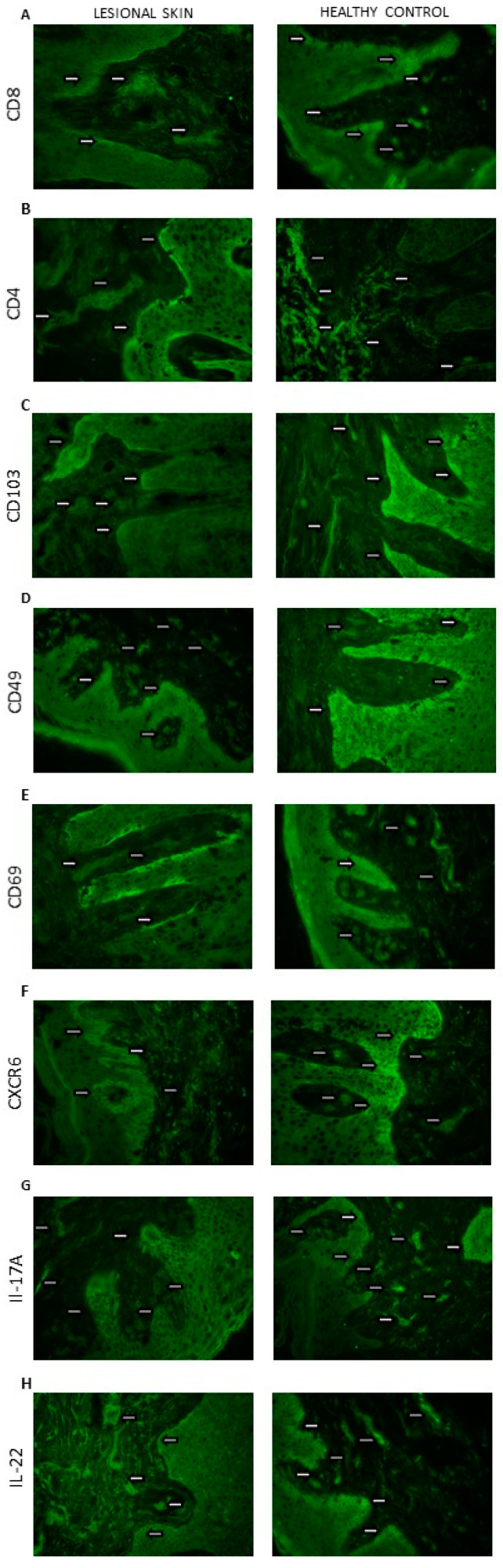
Skin localization of (**A**) CD8; (**B**) CD4; (**C**) CD103; (**D**) CD49; (**E**) CD69; (**F**) CXCR6; (**G**) IL-17A; (**H**) IL-22 proteins (the representative sections): lesional skin (*n* = 32, dermis and epidermis) and healthy control (*n* = 10, dermis and epidermis). The proteins are marked in green (fluorescein). Magnification: 500×.

**Figure 3 ijerph-18-11251-f003:**
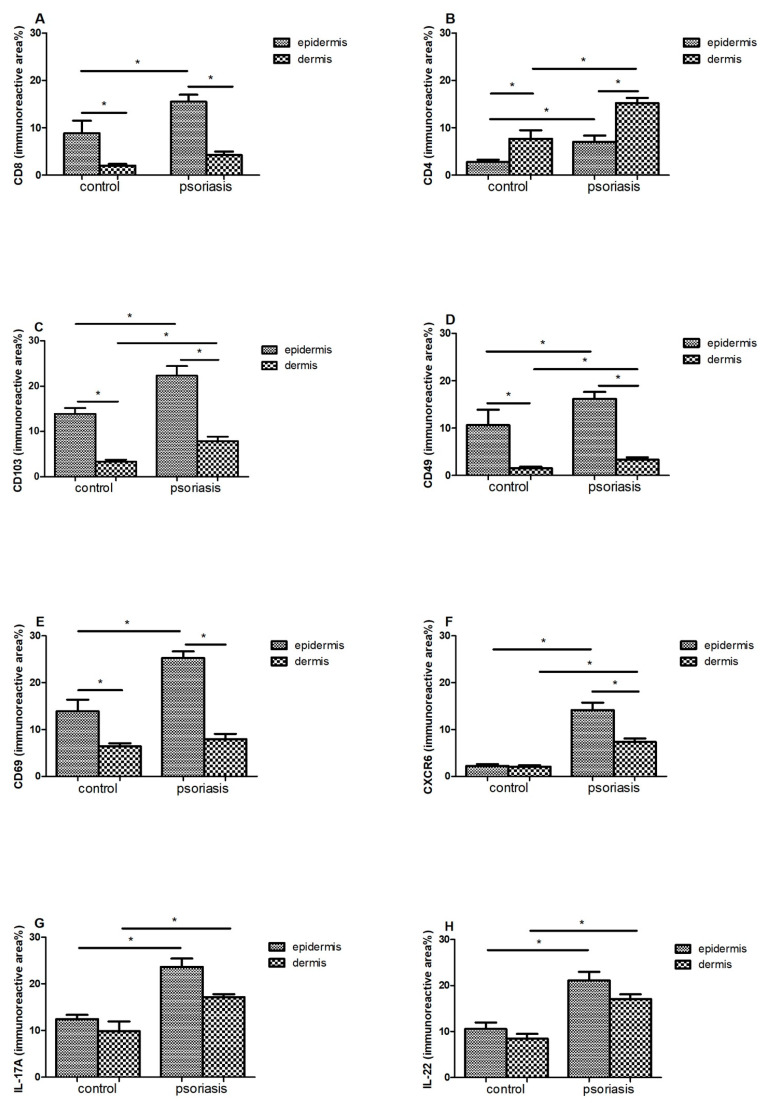
Immunoreactive area (means ± SEM) of (**A**) CD8; (**B**) CD4; (**C**) CD103; (**D**) CD49; (**E**) CD69; (**F**) CXCR6; (**G**) IL-17A; (**H**) IL-22 in lesional skin (*n* = 32, dermis and epidermis) in comparison with healthy control (*n* = 10, dermis and epidermis). Bars with asterisks are significantly different (*p* < 0.05, Mann–Whitney U test).

**Table 1 ijerph-18-11251-t001:** Immunoreactive area (%) TRM markers (X ± SEM) in lesional skin (*n* = 30, dermis and epidermis) in comparison with healthy control (*n* = 10, dermis and epidermis). Asterisks show the statistically significant differences between analyzed groups.

	Healthy Control (*n*: 10)		Lesional Skin (*n*: 32)	
	Epidermis (%)	Dermis (%)	Epidermis (%)	Dermis (%)
**CD8**	8.83 ± 2.68	2.04 ± 0.33	15.49 ± 1.53 *	4.27 ± 0.73
**CD4**	2.77 ± 0.49	7.71 ± 1.80	6.45 ± 0.77 *	15.21 ± 1.11 *
**CD103**	13.88 ± 1.28	3.37 ± 0.42	22.32 ± 2.10 *	7.86 ± 1.02 *
**CD49**	10.66 ± 3.23	1.55 ± 0.34	16.17 ± 1.48 *	3.39 ± 0.42 *
**CD69**	13.89 ± 2.51	6.39 ± 0.69	25.24 ± 1.42 *	7.94 ± 1.13
**CXCR6**	2.24 ± 0.41	2.05 ± 0.33	12.23 ± 1.04 *	7.34 ± 0.78
**IL-17**	12.45 ± 0.94	9.85 ± 2.14	23.63 ± 1.82 *	17.19 ± 0.64 *
**IL-22**	10.54 ± 1.44	8.45 ± 1.03	21.12 ± 1.87 *	17.10 ± 1.01 *

**Table 2 ijerph-18-11251-t002:** Correlations (r, *p* < 0.05) between individual markers in the study group of patients with psoriasis.

Marker (r)	cd8	cd4	cd103	cd49	cd69	cxcr6
CD8	-	−0.37	0.55	0.65	0.64	0.45
CD4	−0.37	-	−0.29	−0.47	−0.46	−0.26
CD103	0.55	−0.29	-	0.64	0.6	0.51
CD49	0.65	−0.47	0.64	-	0.69	0.54
CD69	0.64	−0.46	0.6	0.69	-	0.52
CXCR6	0.45	−0.26	0.51	0.54	0.52	-

**Table 3 ijerph-18-11251-t003:** Correlations (r, *p* < 0.05) between individual markers and the duration of psoriatic lesions.

Marker (%)	Duration of the Psoriasis (years)
CD8	0.45
CD4	0.55
CD103	0.44
CD49	0.57
CD69	0.38
CXCR6	0.59
IL-17A	0.37
IL-22	0.51

**Table 4 ijerph-18-11251-t004:** Correlation coefficient (r) between the analyzed variables: time, intensity, and expression. Statistically significant correlations are marked in italics and bold.

	PASI	BSA	DLQI
time (years)	0.21	0.15	0.29
Cd8 (%)	0.22	0.17	0.33
Cd4 (%)	0.20	0.16	0.27
Cd103 (%)	0.21	0.16	0.31
Cd49 (%)	0.23	0.18	0.28
Cd69 (%)	0.25	0.20	0.29
Cxcr6 (%)	0.18	0.13	0.28
IL-17A (%)	0.20	0.15	0.30
IL-22 (%)	0.24	0.15	0.31
PASI	-	* **0.95** *	* **0.56** *
BSA	* **0.95** *	-	* **0.59** *
DLQI	* **0.56** *	* **0.59** *	-

## Data Availability

Not applicable.

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
