# Peer review of "Assessment of the Tissue Resident Memory Cells in Lesional Skin of Patients with Psoriasis and in Healthy Skin of Healthy Volunteers"

_ijerph, 2021, doi:10.3390/ijerph182111251_

Round 1

Reviewer 1 Report

Marta Kasprowicz-Furmańczyk and colleagues present a highly interesting study on the frequency of tissue resident T cell markers in psoriatic lesions. The manuscript is overall well-written, with ample representation of the data and clear conclusions, although somewhat lacking methodological details.

Major comments:

  1. Statistics: please use “Mann-Whitney U-test” instead of “U Manna-Whitney”.
  2. X +/- SEM is incorrect, please use mean (or median) +/- SEM. Moreover, using standard error of the mean is highly discouraged, standard deviation should be used whenever possible.
  3. Correlation analysis is not described in Methods (Pearson, Spearman or other?). Please use “correlation coefficient” instead of “correlation factor”. Importantly, the threshold of correlation considered significant should also be stated in Methods.
  4. Section 2.2 (goals) is more fitting to a thesis, but should be omitted from a manuscript.
  5. Figures: in boxplots, why is lettering used to indicate statistically significant differences? Letters do not seem to correspond to sample groups. Asterisks and horizontal bars (or better, the actual p-values) are standardly used for labelling.
  6. The correlation matrix in Table 2 presents interesting results indeed; do I see it correctly that all values are significant (p<0.05), even the moderate negative correlations, e.g. CD4-CXCR6? If so, might there be a systematic explanation for this phenomenon?

Minor comments:

  1. English language: spell-checking is highly recommended, e.g. “TRM markers” in most use cases should not be plural, etc.
  2. As the ratio of healthy controls to psoriatic subjects is not ideal (~1:3), the authors should consider providing supplementary information, e.g. the body sites from which samples were taken, as body site heterogeneity is an important confounding factor and might aid interpretation of the results.
  3. While presenting each marker in a separate figure seems logical, the authors should consider grouping the markers into perhaps two figures, as the figure structure is repetitive and it would be more in line with publication standards. Labelling patient / control and the marker in the images themselves would improve readability.

Author Response

Good evening,

Kind regards,

Marta Kasprowicz-Furmańczyk

Reviewer 2 Report

I read with great interest the article titled: Assessment of the TRMs in lesional skin of patients with psori-2 asis and in healthy skin of healthy volunteers.

The article looks great and very detailed in the methods and results, so I have only few suggestions to extend.

TITLE: Please report the meaning of the acronym TRM

ABSTRACT:

TRM are also in charge to the koebner phenomenon.

INTRODUCTION

Please add a sentence that contextualize psoriasis as Th1, Th17, Th22  systemic inflammatory disease [PMID: 29742056.] characterized by several comorbidities encompassing respiratory[10.1155/2018/3140682, 10.1007/s10067-020-05050-2], cardiologic [10.3390/jcm9010186, 10.1016/j.jid.2019.07.727] and gastrointestinal ones [10.1155/2018/3140983].

Please add a paragraph on Koebner and TRM.

METHODS

Please add in the exclusion criteria concurrent dermatoses, specify the diet as omnivores since it is related to psoriasis flares  [10.3390/nu13092934] or addictions [10.3390/jcm8060770].

RESULTS

please divide results into paragraphs with the explanatory headings.

Author Response

(The authors gave the same response as above.)

Reviewer 3 Report

This article addresses the concept of immunological memory in psoriasis by characterizing tissue resident memory cells (TRMs). Using immunohistochemistry (IHC) authors show a vast repertoire of cell surface markers for TRMs in healthy and psoriasis lesional skin. Furthermore, this study presents a positive relation between TRM markers and the duration of skin lesions in psoriasis. There are few points of concern which are listed below,

  1. Several articles already investigated immunological memory in psoriasis by interrogating the role of TRMs and their surface markers. For example, PMID: 28392462, PMID: 26782974, PMID: 30268387, PMID: 29510191, PMID: 28214226 and others. This weakens the strength of the presented article and reduces novelty. What is the strength of the presented results in light of these and other published articles describing the role of TRMs in psoriasis?
  2. Figure 1-8, panel A, small alphabet a, b, c signifies what in the bar graph. Details for this could be included in figure legends.
  3. Figure 1-8, panel B, healthy and lesional skin is not marked. Indicate in the figure, that which image is for healthy skin or psoriasis lesional skin.
  4. Figure 1-8, panel B, an H&E image of the healthy and lesional skin would demarcate the histology of skin in these two conditions, as in the present form it is difficult to visualize the skin histology.
  5. In general human skin show green auto-fluorescence, figure 1-8 panel B, all the interrogated TRM cell surface markers are stained with green (fluorescein), which makes it difficult to distinguish protein signals from skin auto-fluorescence. Although there are arrows marking positive stain, it is difficult to visualize with the green background. All the IHC should be repeated and develop with chromogenic method or with another secondary fluorescence antibody (not green).
  6. There are only 19 references given for this study, as this is a full research article, cite more relevant references.

Author Response

(The authors gave the same response as above.)

Round 2

Reviewer 3 Report

This is a straightforward study presenting evidence about relationship between amount of TRMs present in the skin of patients with psoriasis with the duration of the disease and its severity. The revised manuscript presents findings in a more comprehensive manner. Authors have provided an adequate response to almost all raised concerns. I still have one comment/suggestion which would be beneficial for the readers of this article and would add to the quality of the depicted work.

I still think that an H&E stain would nicely demarcate the histology of the tissue section used. Also was there any nuclear stain used for the presented IHCs? If material is limiting for the H&E staining, authors should at least show the isotype control/no primary antibody control for all the IHC stains. I am assuming at least one of these control was used at the time of IHC. This would deduce the specificity of the observed staining patterns of the interrogated proteins in this article.

Author Response

Good evening,

Thank you very much for your time and further suggestions. As suggested, we tried to include the relevant figures in the article (Fig. 2). We hope that the added fragment will meet your expectations.

Kind regards,

Marta Kasprowicz-Furmańczyk

This manuscript is a resubmission of an earlier submission. The following is a list of the peer review reports and author responses from that submission.